# Genomic Enhancers in Brain Health and Disease

**DOI:** 10.3390/genes10010043

**Published:** 2019-01-14

**Authors:** Nancy V. N. Carullo, Jeremy J. Day

**Affiliations:** Department of Neurobiology, University of Alabama at Birmingham, Birmingham, AL 35294, USA; ngallus@uab.edu

**Keywords:** enhancer, transcription, eRNA, neuron, gene regulation, epigenetic

## Abstract

Enhancers are non-coding DNA elements that function in *cis* to regulate transcription from nearby genes. Through direct interactions with gene promoters, enhancers give rise to spatially and temporally precise gene expression profiles in distinct cell or tissue types. In the brain, the accurate regulation of these intricate expression programs across different neuronal classes gives rise to an incredible cellular and functional diversity. Newly developed technologies have recently allowed more accurate enhancer mapping and more sophisticated enhancer manipulation, producing rapid progress in our understanding of enhancer biology. Furthermore, identification of disease-linked genetic variation in enhancer regions has highlighted the potential influence of enhancers in brain health and disease. This review outlines the key role of enhancers as transcriptional regulators, reviews the current understanding of enhancer regulation in neuronal development, function and dysfunction and provides our thoughts on how enhancers can be targeted for technological and therapeutic goals.

## 1. Introduction

Complex and precise spatiotemporal gene expression patterns orchestrate the wide diversity of cell fate and function. Non-coding DNA elements, such as enhancers, regulate these intricate expression programs at any given time to ensure proper development and function of specific tissues and cell types (Figure 1). Enhancer elements have been studied for over three decades and while these studies have provided tremendous insight into gene regulation and enhancer function, there are still many aspects that remain unknown. Initial studies in *Xenopus* oocytes discovered a sequence upstream of the *H2A* gene that could modulate transcription levels in an orientation-independent fashion. Additionally, deletion of these elements resulted in lower expression levels [1], suggesting that this sequence “enhanced” expression of the downstream gene. However, the term “enhancer” was used for the first time to describe a 72-bp tandem repeat upstream of viral *SV40* gene. This *cis*-acting DNA sequence was found to modulate and increase transcription of a distal gene more than 200-fold, independent of its orientation and location relative to the target gene. Several studies of the viral *SV40* enhancer identified many additional characteristic enhancer features, such as DNase hypersensitivity (indicative of open chromatin) and transcription factor (TF) binding [2,3,4,5,6]. The first studies to provide evidence for cell type specificity of enhancers investigated the mammalian immunoglobin (Ig) heavy chain gene in various cell lines and found that the Ig enhancer was only active in lymphocytes [7].

Enhancers are probably best understood in murine embryonic stem cells and the context of cellular development, followed by cancer as a large portion of research has focused on these areas. Nevertheless, the brain provides a unique window to explore how enhancers contribute to cell fate and function and also to understand how enhancers can be dysregulated in disease. In the brain, enhancers help to ensure cell- and tissue-specific gene expression profiles, defining which genes are active during neuronal specification and which genes remain accessible in adult neurons. In adult neurons, the storage of information requires the dynamic regulation of gene expression patterns to allow for neuronal plasticity, learning and memory formation and long-term adaptations of behavior to an environment. Furthermore, a subset of enhancers shows activity-dependent induction in response to neuronal activation and behavioral experience, suggesting a regulatory role in transcription and brain function.

A significant fraction of non-coding genomic space is dedicated to enhancers, with current estimates suggesting that vertebrate genomes may encode tens of thousands or even millions of active enhancers [8,9,10,11]. It is hence not surprising that the majority of disease-associated single nucleotide polymorphisms (SNPs) fall into non-coding regions of the genome and that an increasing number of these SNPs have been linked to enhancer function [9,12,13,14,15,16,17,18]. Together with emerging roles for enhancer biology in developmental programming and experience-dependent regulation of neuronal systems, these findings highlight the need for more research to understand how enhancers contribute to brain function and disease. While there are other excellent reviews on the general characterization of enhancers [19,20] and neuronal chromatin remodeling [21], this review will focus on the role of enhancers in the nervous system and discuss aspects of enhancer function in heathy and functioning brain, as well as how dysregulation of enhancers may contribute to developmental, neuropsychiatric and neurological disorders.

## 2. General Enhancer Properties and Functions 

### 2.1. Enhancer Properties and Molecular Interactions

Enhancers and enhancer states are characterized by
(1)an open chromatin structure, indicated by DNase hypersensitivity,(2)specific epigenetic modifications,(3)defined molecular interactions, and(4)bi-directional transcription starting in the center of the enhancer leading outward on opposite stands [20,22,23,24,25,26,27,28,29].

This section will explore the prominent features of enhancers, contrast enhancers to other *cis* regulatory elements and discuss potential functions of enhancer clustering and phase separation in gene regulation.

### 2.2. Enhancer versus Promoters

Many *cis* regulatory elements share architectural similarities and enhancers provide a key example of this. Distal enhancers and gene-proximal promoters are defined by many structural and functional features [19,30]. Traditionally, promoters are defined as DNA stretches that initiate transcription from a nearby transcription start site (TSS) [31]. The eukaryotic RNA polymerase 2 promoter contains the TATA-box, initiator, TFIIB recognition element and a downstream core promoter. However, enhancers can also exhibit “promoter-like” attributes. For example, enhancers can contain clusters of TF motifs, although enhancers and promoters tend to be biased towards occupancy by different TFs [32]. Likewise, enhancers can have promoter activities that result in transcription initiation [32,33]. Building on previous observations of promoters with enhancer function [32,34,35,36], Dao et al. used a genome wide approach (CapStarr-Seq) to characterize genomic regions that can act as a promoters and drive local transcription or act as enhancers and increase distal gene transcription. While enhancers and promoters showed distinct genomic and epigenetic characteristics, they found a significant fraction of core promoters harboring enhancer activity in both tested cell types. They further showed that the *FAF2* promoter interacts with three other promoters as an enhancer and that deletion of this promoter decreased expression of all three target genes [37]. More recent work has implied that enhancers and promoters can be repurposed and that an element can function as enhancer in one species and its orthologue can act as a promoter in another (related) species [30]. Importantly, both structures communicate with each other and work together to orchestrate and initiate transcription.

### 2.3. Enhancer Looping

One common way in which enhancers regulate gene expression occurs via direct interaction with promoters through physical enhancer-promoter loops. This enhancer looping mechanism allows distal enhancers to act on genes that are thousands (or in some cases, millions) of bases upstream or downstream of the gene locus and provides another layer of control over dynamic transcriptional responses to internal and external stimuli.

Several factors regulate and stabilize this looping process, including TFs like YY1, the CCCTC-binding factor CTCF and the Mediator/Cohesin complex. CTCF, a TF often found at transactivation domain (TAD) boundaries, is thought to stabilize loop anchors [20]. CTCF further positions the Cohesin complex, which forms a ring-like structure. This structure can wrap around two DNA segments to keep them in close proximity, providing physical means to connect enhancers and promoters [27]. YY1 is an ubiquitously expressed TF that binds hypomethylated DNA and generally occupies enhancers and promoters. Interestingly, YY1 interacts with CTCF to facilitate enhancer-promoter looping and loss of YY1 disrupts enhancer function and transcriptional regulation [28,38]. Finally, Mediator is a multiprotein complex of 30 subunits that is crucial for the expression of almost all protein coding genes. This complex coordinates signals at enhancers and recruits RNA polymerase 2 (RNAP2) along with a number of TFs, facilitating the formation of the preinitiation complex. Mediator has further been shown to be involved in enhancer-promoter loop formation through direct interactions with the Cohesin complex [29,39,40].

### 2.4. Enhancer Chromatin Architecture

Enhancer regions are typically more conserved across species than surrounding regions in the genome. In fact, initial genome-wide enhancer screens used sequence conservation as an indicator for enhancers [41]. While developmental enhancers showed an enrichment in DNA sequence conservation, screening of hundreds of conserved regions in the genome found that a significant proportion possessed enhancer-like activity in reporter assays [42,43].

As enhancers are characterized by open chromatin, techniques based on chromatin accessibility such as mapping of DNase hypersensitive sites or Assay for Transposase Accessible Chromatin Sequencing (ATAC-seq) are often used to identify enhancers across the genome [22,44]. This information is often used in combination with histone modifications to identify enhancer regions and further define their functional states. For example, active enhancers are typically enriched in H3K27ac [45], whereas poised enhancers often exhibit H3K27me3—a modification associated with the Polyclomb repressive complex [25]. While promoters often show low H3K4me1 versus H3K4me3 ratios, enhancers show the opposite pattern and present with high levels of H3K4me1/me2 and low levels of H3K4me3 [25,45]. Together, these characteristics are commonly used to identify enhancers and to determine their state of activity (e.g., active vs. poised) on a genome-wide scale.

Interestingly, histones in enhancer regions show an enrichment of H2A.Z and H3.3 histone variants. These variants are thought to be less stable (and easier to displace) than traditional histones, allowing for more dynamic mobility and easier access for TFs to bind [46,47]. In addition to histone modifications and TF binding, active enhancers are further characterized by DNA hypomethylation, as well as active bi-directional transcription that yields non-coding enhancer RNA (eRNA). The last decade has yielded exciting insights into the role of these non-coding transcripts, which is discussed later in this review.

### 2.5. Enhancer Interactions

Enhancer regions are subject to dynamic epigenetic modifications and binding of chromatin remodelers such as histone acetyltransferases (HATs). CBP and p300, two closely related HATs, target histone tails as well as transcription machinery including RNAP2 in TAD domains. While histone acetylation creates a more accessible and permissive chromatin structure at enhancers and improves TF recruitment, RNAP2 acetylation promotes its release into the gene body [48,49,50]. Nevertheless, H3K27ac and CBP/p300 binding are hallmarks of active enhancers [49,51] and H3K27ac is commonly used as a surrogate measure of enhancer activity.

The histone methyltransferases Kmt7 (also known as Set7) has been found to bind enhancers in muscle tissue in addition to HATs. Set7 binding increases H3K4me1 at enhancers and is also associated with increased target gene expression [27]. Other groups have made similar discoveries in identifying SET-domain containing proteins Mll3 and Mll4 as major contributors to enhancer H3K4 methylation [52,53]. Together, these studies demonstrate the critical importance of histone methylation at enhancers and further support the role of SET-domain proteins in establishing this methylation.

The binding of epigenetic modifiers is also crucial for TF function and the ability of enhancers to drive gene expression. Enhancer activation is thought to require the binding of several TFs, often including lineage-specific and motif-dependent factors that respond to signaling cascades to integrate extrinsic and intrinsic signals. These early TFs then prime the enhancer region and reinforce binding of additional TFs and coactivators. The combinatorial binding of different TFs and coactivators then gives rise to discrete expression patterns across different tissues and cell types.

Extensive investigation of developmentally regulated enhancers in stem cells has shown how master TFs like Oct4, Sox2 and Nanog can identify or predict enhancer activity and that TF function often depends on the binding of coactivators that harbor chromatin remodeler activity or mediate long-range interactions [54,55]. In pluripotent embryonic stem cells, these master TFs bind super enhancers and recruit Mediator to activate their target genes. Intriguingly, cell type specific TF binding to enhancers of more differentiated cells often regulates cell identity genes [54].

Recent reports have suggested that a subset of TFs is biased towards different regulatory elements like enhancers or promoters. AP1, a TF composed of FOS/JUN heterodimers, is biased towards enhancer elements, whereas EGR1 and SP1 are biased towards promoters and promoter activity. In contrast, a subset of TFs, including RFX, is capable of generating both enhancer and promoter activities [32].

The key role of TF binding in enhancer activation is not limited to developmental processes, as a striking subset of enhancers show activity-dependent TF binding. For example, CBP and Npas4 were found to bind enhancers surrounding the *Fos* gene in response to neuronal depolarization [56] and many activity-regulated enhancers are characterized by increases in H3K27 acetylation [57]. Surprisingly, a subset of neuronal activity-dependent enhancers requires Fos binding, which is consistent with the bias of the AP1 complex towards enhancer elements in the genome [58].

### 2.6. Super Enhancers

The term super enhancers (SEs) refers to clusters of enhancers that are characterized by increased size, abundant TF and Mediator complex occupation and increased capability to induce gene expression compared to “typical” individual enhancer loci. SEs dictate expression patterns of entire groups of functionally related genes. Whyte et al. identified SEs that regulate cell identity genes in mouse myotubes, T helper cells and macrophages [54]. On a genome-wide scale, SE-regulated genes tend to require high levels of coordination and fine tuning of their expression. They regulate expression in a particular cell and tissue type, during specific developmental stages or in response to unique cellular stimuli. Interestingly, mutations that fall into or impair SEs have been linked to various diseases such as various forms of cancer, diabetes and also Alzheimer’s disease (discussed later in this review in Section 3.3, Section 3.4, Section 3.5 and Section 3.6) [9,54].

### 2.7. Enhancer Transcription

Another hallmark of active enhancers is bi-directional transcription, which originates in the center of the enhancer and produces RNA transcripts from both DNA strands in opposing directions. The act of transcription itself has been suggested to keep the enhancer locus in a more permissive and active state, accessible for TFs, epigenetic modifiers and transcription machinery. Nevertheless, these transcripts arising from enhancers (so called enhancer RNAs or eRNAs), were initially thought to result from their close proximity to RNAP2 and actively transcribed genes. Thus, eRNAs were not initially included in models of enhancer function and their specific roles have remained controversial. However, over the last decade striking evidence for the functional relevance of eRNAs has emerged. Several studies have demonstrated that eRNAs interact with several key molecules in enhancer function including the Mediator complex, transcriptional repressors and activators, as well as epigenetic machinery [59,60,61,62,63]. The role of eRNAs as a functional unit of enhancers is an exciting and widely understudied field that will help understand the complex regulation of and by enhancers.

### 2.8. Enhancers and Phase Separation

Phase separation describes molecular condensates within eukaryotic cells that create compartments of high protein and nucleic acid density, such as the nucleolus, nuclear speckles, stress granules and many other compartments. The formation and specific composition of phase-separated compartments is tightly regulated to provide the opportunity for controlled environments optimal for specific reactions. The investigation of phase separation has recently gained a lot of momentum, as striking discoveries have demonstrated that the components of these condensates can change over time depending on the state of cell cycle or in response to certain stimuli (reviewed in Reference [64]). Intriguingly, enhancers have been linked to phase separation. A recent review by Hnisz et al. discusses the phase separation model for transcriptional regulation that allows linking of all components needed for transcriptional induction including enhancers, promoters, TFs, epigenetic modifiers and transcription machinery [65]. In this model, phase separation mediates the accumulation of high concentrations of these components in a confined space that allows for necessary chemical modification of interacting components. Furthermore, this model describes characteristics of enhancers and SEs based on three main parameters:(1)the number of participating components, (2)all possible chemical modifications of their residues,(3)and the rate at which these modifications are facilitated or inhibited.

Specifically, SEs are suggested to rely on high levels of such cooperativity between numerous interaction partners, as they typically accumulate higher numbers of interacting molecules than traditional enhancers. Overall, this model addresses how phase separation could play a role in transcriptional control and explains how it might contribute to previously observed characteristics of SEs.

Phase separation as a mechanism for enhancer function is particularly interesting as it would enable concentration of a variety of components necessary for regulated gene transcription. Sabari et al. were the first to demonstrate the participation of SEs in phase separation by showing that coactivators Mediator and BRD4 (a member of the BET family), associate with SEs in nuclear condensates. Disruption of the condensates by 1,6-Hexanediol resulted in loss of Mediator and BRD4 binding at SEs and loss of RNAP2 occupancy at SEs and their target genes. This work suggests a role of coactivators in the compartmentalization and local concentration of SEs and transcription machinery [66], perhaps via disordered domains in these proteins.

## 3. Brain Specific Enhancer Functions 

### 3.1. Enhancers in Neurodevelopment

Enhancers control complex spatiotemporal expression programs throughout development and into adulthood and so initial studies on neuronal enhancers focused on identification of enhancer loci in the developing CNS. In a pioneering study, Nord et al. measured H3K27 acetylation with Chromatin immunoprecipitation sequencing (ChIP-Seq) as a readout of active enhancer loci across seven developmental stages ranging from embryonic day 11.5 to postnatal day 56 to identify enhancer activity profiles in three different tissues including the mouse forebrain [14,67]. This work revealed tissue and temporally specific enhancer activity profiles in which 85% of the enhancers only showed active histone marks during specific developmental stages. To experimentally test these H3K27ac ChIP-seq findings in vivo, Nord et al. capitalized on transgenic mice in which the LacZ reporter was driven by a minimal promoter coupled with regions identified as active enhancers [67]. Their results confirmed not only that these enhancer sequences can drive tissue specific gene expression but also demonstrated that the dynamic H3K27 acetylation changes throughout development correspond to gene expression changes (measured by LacZ reporter expression). Most importantly, these regions were also associated with critical tissue-specific biological processes. Motifs for TFs that control neuronal differentiation (such as Lhx3) were enriched in early active enhancers, whereas motifs for TFs known to regulate synaptic transmission, cognition, learning and memory and neurodegeneration were enriched in delayed active enhancers [9,54]. Finally, this approach led to the discovery that the lead SNP for depression and alcohol dependence fell into a region with enhancer activity in the forebrain [14,67]. Together, these findings highlight the advantages of H3K27ac datasets to reliably identify active enhancers across developmental stages as well as the implications of enhancer function in disease development.

Building on these findings, subsequent studies have investigated developmental changes in chromatin accessibility and enhancer activity in specific cell types in the cerebellum [59,60,61,62,63]. This work revealed an enrichment for TF binding motifs in developmentally regulated open chromatin (DNase hypersensitive sites) between postnatal day 7 and 69. Along with TFs that are important for cerebellar granule cell differentiation (MEF2 and NF1 families) Frank et al. also found an enrichment for the Zic motif, a TF which has previously been associated with cerebellar development disorders [14,67]. Zic binding promotes maturation of postmitotic cerebellar granule neurons while preventing premature differentiation of progenitor cells. Overall, this work demonstrates how chromatin changes at enhancer loci regulate accessibility to TFs and thereby gene expression programs in neuronal subtypes across different time points (Figure 2a).

### 3.2. Cell Type and Brain Region Specific Enhancers

While initial studies of enhancers in the 1980s established a cell type specific role for enhancers [64], later genome-wide studies identified specific enhancer subsets that regulate expression patterns in the nervous system. Nord et al. not only showed temporally precise windows of enhancer activity but also that each tested tissue was characterized by a unique enhancer activity signature [66]. There is accumulating evidence for cell-type specificity of enhancers that are characteristic for neural stem cells and neurons, as well as overall organ/tissue specific enhancers that are active in the brain but not elsewhere. Andersson et al. used cap analysis of gene expression (CAGE) to identify active enhancers based on their bidirectional transcription in a number of cell and tissue types [66]. Next, this dataset was integrated with datasets for DNA methylation and histone modifications (ChIP-Seq for H3K27ac and H3K4me1) to identify active enhancers in five primary blood cell types. Several of the identified enhancers were then validated in vivo in transgenic zebrafish embryos. Overall, these results show that many enhancers are active across many cell types but that there are subsets of enhancers specific for a certain tissue (brain, blood, liver and testis) or cell type (immune cells, neurons, neural stem cells and hepatocytes). These subsets of cell type and tissue specific enhancers were found to be enriched in binding motifs for the same key regulators including RFX and SOX in neurons. Similarly, ChIP-Seq analysis of microdissected subregions of the adult mouse cortex revealed novel region-specific enhancers. While most of these enhancers were active in at least two subregions, about a third of the identified enhancers were subregion-specific [68]. Further analysis revealed that within these region-specific enhancers, a subgroup exhibits specificity for a certain cell type within that region. Interestingly, these enhancers drive transgene expression in the same subregion in a cell-type selective manner, confirming their specificity while also providing the opportunity for exciting new tools capable of highly specific transgene expression. New transgenic enhancer models can build on this model and provide higher resolution than current approaches to drive cell-type and region-specific transgene expression. These models can be combined with inducible systems (such as a tetracycline-dependent system) to allow for extremely fine-tuned cell type specific, spatio-temporal expression profiles [69].

More recently, examination of enhancer activity states has been extended to postmortem brain samples to begin defining cell class and cell type dependent enhancer programs in the human brain. Fullard et al. investigated *cis*-regulatory elements, including promoters and enhancers and created a map of chromatin accessibility in neuronal and non-neuronal nuclei across 14 brain regions of human postmortem tissue. This study revealed differences between neuronal and non-neuronal cell types, as well as differences among neurons from different brain regions [44]. Similar approaches have extended this work as part of a broad characterization of non-coding regions (including enhancers) [70]. This work compared non-coding elements between dopamine neurons, pyramidal neurons and non-neuronal cells. A portion of the transcribed non-coding elements coincided with other enhancer characteristics such as DNase I hypersensitivity, certain histone modifications (high H3K27ac, H3K4me and low H3K3me3), CAGE-defined enhancers, high sequence conservation or binding sites for p300 and TFs. Remarkably, this study found that a large portion of detected enhancers were exclusively expressed in their respective cell type, again highlighting the specificity of enhancers (Figure 2a).

How is enhancer activity regulated in a spatiotemporal precise manner? One route of regulation is enhancer accessibility to interaction partners, which is determined by epigenetic modifications at enhancers. For example, a recent study found that the epigenetic state of enhancers affects gene expression and neuronal migration patterns. The histone methyltransferase PRDM16 is a crucial regulator of the epigenetic state of enhancers in the embryonic cortex and is thought to deposit H3K9me1 and H3K4me1 [71]. This study suggests that PRDM16 binds to active enhancers (measured by H3K27ac) and found an overlapping enrichment in H3K4me1 which is typically found at active and poised enhancers. In addition to regulating genes involved in cortical neurogenesis, PDRM16 also represses the expression of migration genes in radial glia cells to direct proper positioning of cortical neurons. This enhancer mediated expression profile ensures generation of the accurate number of cortical neurons and their correct positioning in the upper layer of the cortex [71].

Once enhancers are accessible, cell identity is mediated by coordinated binding of specific TFs to specific enhancers. Rhee et al. characterized how initial TF binding dictates differentiation and cell identity and how the interplay between different TFs ensures stable expression patterns in mature motor neurons [72]. More specifically, they found that the TF Isl-1 engages with either Onecut1 or Lhx3 depending on the developmental stage. While Isl-1 (along with histone acetyltransferases) is initially bound to Lhx3, it is released from motor neuron enhancers and recruited to Onecut1 clusters in later developmental stages of maturing motor neurons [72]. These findings display how coordinated and temporally specific binding of TFs like Onecut1 and Lhx3 to integrator TFs such as IsI-1 results in intricate expression patterns. Understanding how different cell types and even subtypes in the brain are regulated in this way will enable us to better connect how synaptic signals are integrated into transcriptional responses to different stimuli with cell type specific resolution (Figure 2a).

### 3.3. Experience—And Activity—Dependent Enhancers in the Brain

In the context of brain function, one important question emerges: how do enhancers regulate genes in response to neuronal stimulation? Activity-dependent gene regulation is a hallmark of key neuronal phenomena, including long-term potentiation, homeostatic plasticity, long-term memory formation and adaptive behavior. Therefore, a central focus of ongoing attempts to understand enhancer function in the central nervous system has utilized activity-responsive enhancers found near genes that are rapidly controlled by neuronal depolarization and other stimuli.

Addressing this question, a pioneering study by Kim et al. discovered ~12,000 activity-regulated enhancers in murine neurons. Enhancer activation was accompanied by stimulus-dependent CBP recruitment to enhancers near activity-regulated genes (e.g., *Fos*, *Rgs2* and *Nr4a2*). Activity-dependent recruitment of Creb, Mef2, Npas4 and Fos to enhancer regions further demonstrated the importance of this activity-dependent mechanism for enhancer regulation [73,74]. Not only did enhancers respond to stimulation but their responses were specific to the received stimuli, potentially providing an opportunity to fine tune transcriptional responses to various forms of neuronal activation [26]. This has been suggested to be the underlying mechanism of transcriptional plasticity in response to experience [11].

The best characterized activity responsive enhancers are located around the *Fos* gene. The five enhancers surrounding the murine *Fos* locus have been shown to respond to neuronal activity [26,57,58] and show different activation patterns (measured by eRNA induction) in response to neuronal depolarization (KCl), neurotrophic factor stimulation (BDNF) or cyclic AMP signaling (Forskolin) treatments. While all three treatments increased mRNA expression, each of the surrounding enhancers showed different levels of induction to the respective stimulus. The same stimuli evoked enhancer activation in a luciferase reporter assay with a minimal *Fos* promoter and the respective *Fos* enhancers. These authors further confirmed that in vivo, different stimuli (kainic acid injections or sensory stimulation of the visual cortex with light) result in different combinations of enhancer activation. Together, these findings highlight how combinatorial enhancer activation can distinguish between different stimuli and potentially fine tune and adjust the transcriptional response to a certain stimulus [56]. Importantly, this demonstrates that enhancers can alternate or compete to regulate the same gene but can also act in combination to promote additive or synergistic effects (Figure 2b). In other cases, one enhancer or group of enhancers might regulate several related genes either simultaneously or selectively (Figure 2c). Additionally, entire networks of genes may be linked by SE complexes that help to maintain expression of programs that define neuronal function (Figure 2c).

Another well studied activity dependent enhancer is located upstream of the murine *Arc* gene. This immediate early gene encodes a cytoskeleton protein crucial for synaptic plasticity and has been shown to underlie enhancer mediated regulation [62,75]. Neuronal activity induces enhancer-promoter interactions at this locus and eRNA induction is crucial for the release of the negative regulator of transcription elongation (NELF) and RNAP2 to initiate gene transcription [62]. Similarly, Telese et al. showed that Reelin, an extracellular matrix protein that modulates synaptic plasticity and long term potentiation [76,77], induces a subset of neuronal enhancers when bound to its receptor (LRP8) [78]. Reelin-activated enhancers thereby translate synaptic inputs into transcriptional programs important for synaptic plasticity, suggesting that activation of these enhancers might also be fundamental for learning and memory formation.

Together with their cell type specific nature, enhancers’ ability to respond to neuronal activity is extremely important to understand how different cells in circuit or brain region function on a cellular as well as on a network level (Figure 1 and Figure 2). This understanding can help us understand how different stimuli can influence gene expression, how gene expression can drive neuronal physiology and how this interplay can ultimately influence behavior in developing and adult animals. Other excellent reviews have focused on the role of enhancers in neuronal activity in more detail [11,79,80].

### 3.4. Enhancer Dysregulation in Brain Disease

Numerous studies have linked neurological diseases to mutations and variants in DNA. While in some cases a specific gene mutation can be identified as the cause for a disease, for many diseases of the brain this is not the case. However, a large number of sequence variants have been linked to such diseases via genome-wide association studies. Interestingly, the majority of disease linked SNPs fall into non-coding regions and are increasingly linked to enhancer function. As normal enhancer activity requires a complex sequence of interaction partner binding and chromatin remodeling to induce transcription (Figure 3a), the logical conclusion is that mutations in such interaction partners (Figure 3b) or different enhancer variants (Figure 3c,d) likely induce different levels of expression at their target genes and might predispose or even contribute to disease development [9,13,70].

Genetic mutations could interfere with enhancer function and disrupt transcriptional regulation in various ways (Figure 3). First, mutations linked to enhancer regions could lead to aberrant eRNA expression (e.g., incorrect levels or improperly folded eRNAs) or even cause a complete loss of eRNA transcription. Secondly, mutations in TFs and epigenetic modifiers that are normally recruited to enhancers could lead to significant dysregulation of enhancer function across the genome (Figure 3b). Finally, enhancer mutations could cause loss of binding motifs for key regulators of enhancer looping or regulation by TFs, which would impair interactions with promoter regions (Figure 3c) or even lead to increased interactions and enhancer function resulting in over-activation of the target gene (Figure 3d). The following subsections will discuss these possibilities in more detail. 

### 3.5. Mutations of Enhancer Loci

Neurodevelopmental disorders—Prevalent neurodevelopmental disorders have been linked directly to polymorphisms in enhancer regions. Autism Spectrum Disorder (ASD) presents with a wide range of clinical symptoms and numerous mutations and genetic variants have been linked to increased ASD susceptibility. The 5p14.1 locus is a region between the cadherin *CDH10* and *CDH9* genes and a prominent locus for GWAS hits for ASD linked variants. This ASD-associated locus exhibits enhancer activity that regulates expression in cortical layer II/III, striatal and cerebellar neurons in human BAC2 transgenic mice [16].

A number of diseases have been associated with short tandem repeats [81] and these also have the potential to alter enhancer function. One such repeat has been linked to Fragile X syndrome, a genetic neurodevelopmental disorder characterized by intellectual and learning disabilities. Sun et al. discovered that many disease-associated repeats fall into chromatin domain boundaries [81]. In Fragile X syndrome, a CGG triplet repeat disrupts TAD boundaries around the *FMR1* gene, resulting in impaired CTCF binding. Aberrant CTCF binding could disturb enhancer looping to target promoters and result in altered gene expression levels. Neurological diseases—Even though enhancers are typically associated with development, enhancer dysregulation does not only lead to neurodevelopmental problems but has also been implicated in neurodegenerative and psychiatric diseases, as well as in mental health. Alzheimer’s disease (AD) is a devastating progressive neurodegenerative disease characterized by dementia and loss of cognitive function. Various studies have identified and linked mutations and sequence variants to AD, including *APP*, *APOE4* and *BIN1* [9,82,83]. Hnisz and colleagues identified 5 SNPs linked to AD in SE regions of the brain. Two of these SNPs fell into a SE that regulates *BIN1* expression [9,83]. Another recent study associated the *PM20D1* locus with AD risk [84]. The *PM20D1* locus interacts via CTCF-mediated chromatin loops with an AD-associated haplotype that shows enhancer-like characteristics. This region displayed an enrichment for epigenetic enhancer marks and increased expression in a luciferase assay and the interaction with *PM20DI* was haplotype dependent. Interestingly, *PM20DI* expression increased in response to neurotoxicity and overexpression alleviated cell death, decreased Aβ levels and improved cognitive function. Therefore, risk-haplotype carriers might be left with an impaired cellular defense in AD as a result of decreased *PDM20DI* expression. Together these studies highlight the potential impact of enhancer function on AD predisposition and manifestation.

Parkinson’s disease (PD) is one of the most common neurodegenerative disorders and is characterized by late disease onset and complex interactions between genetic and environmental risk factors. The *SNCA* gene was identified as one of the strongest risk loci associated with sporadic PD and mutations of the gene have been shown to cause familiar PD [85,86]. Intriguingly, Soldner et al. identified a PD-associated SNP that falls into an enhancer element in intron-4 of the alpha-synuclein gene *SNCA* in patient derived pluripotent stem cells [87]. Alpha-synuclein is a key factor in PD development and pathogenesis and this work suggests that an enhancer region variant is accompanied by disrupted TF binding and altered *SNCA* expression. This study demonstrates how sequence variants in enhancers can impair enhancer functions such as proper TF binding and thereby promote disease-related expression imbalances.

Psychiatric disorders—Schizophrenia (SZ) is one of the most common psychiatric disorders characterized by positive, negative and cognitive symptoms and a complex interplay between genetic and environmental risk factors. There is increasing evidence for an enrichment of enhancers and promoters in SZ-linked gene variants and several SZ associated SNPs have been shown to fall into non-coding regions with enhancer activity [15,70,88,89]. Variants in or near the L-type calcium channel subunit CANCA1C have been shown to fall into enhancer regions and affect their interactions and functions [15,70,88,89]. These enhancer SNPs lead to transcriptional dysregulation, possibly contributing to SZ manifestation. This new layer of gene regulation could help untangle the genetic and clinical heterogeneity of SZ.

Alcohol dependence and depression are psychiatric disorders that often occur together. A number of genetic variants likely influence the development of both disorders [14]. Two SNPs that were associated with depression have been shown to fall into the *GAL* enhancer. As proper *GAL* expression is required for food and alcohol intake as well as the regulation of mood, these SNPs could also have implications in alcoholism and obesity. Interestingly, the more conserved sequence shows strong enhancer activity while the less conserved and disease-linked variant was associated with decreased gene induction capacity [12]. Thus, enhancer-mediated gene induction could be directly contributing to the phenotypes observed in the different diseases.

As we continue to study disease associated SNPs, the importance of enhancer function becomes abundantly clear. Future work will not only identify other disease-linked regions with enhancer activity but also help understand the underlying mechanisms of enhancer function and their contribution to disease manifestation.

### 3.6. Mutations of Interaction Partners

In addition to mutations in enhancer elements directly, mutations in interaction partners can also obstruct enhancer function and result in disease-related expression patterns. For example, mutations of the Mediator complex, which is crucial for enhancer looping, have been associated with a number of neurological and neurodevelopmental disorders [90,91,92,93,94,95,96]. Mutations in different Mediator complex subunits like MED12 and MED23 disrupt transcriptional regulation of immediate-early genes and have been linked to intellectual disabilities. Similar developmental delays and defects develop as a result of mutations of Cohesin or the Cohesin loading protein NIBL. These disorders are also referred to as Cohesinopathies [29,97,98,99,100]. Likewise, haploinsufficiency of YY1 has recently been shown to cause chromatin and transcriptional dysregulation and this is likewise associated with intellectual disabilities [28]. The histone acetyltransferases CBP and p300 function as transcriptional coactivators in the regulation of enhancer function and gene expression. Mutations in either CBP or p300 have been shown to cause the congenital disorder Rubinstein-Taybi syndrome [101], which is associated with neurodevelopmental and learning disabilities. Thus, a common feature of enhancer-linked proteins is that mutations in these proteins give rise to cognitive disorders (see Figure 3b). All of these mutations are thought to disrupt chromatin structure and enhancer-mediated transcriptional regulation that ultimately results in expression changes that underlie these intellectual disabilities.

### 3.7. Future Directions

With growing understanding of the importance of enhancers in brain health and disease, future research will need to elucidate the mechanisms and dynamics of enhancer activity in response to neuronal stimulation. While the ability of enhancers to respond to stimuli such as neuronal depolarization has been appreciated for nearly a decade, the role of different enhancers in regulating the response of the same gene to different stimuli is much less clear. This activity could potentially provide a key mechanism by which a specific context or experience can be translated and integrated into specific transcriptional response patterns. Thus, a critical area of focus should be to unravel how enhancer activation and inactivation is mediated in different brain regions and in response to different intrinsic and extrinsic stimuli. This knowledge will help explain how enhancers regulate and fine tune transcriptional responses in a specific context and how the disruption of enhancer activity contributes to disease formation.

Likewise, even though we know several key events that regulate enhancer activation (such as binding of key TFs and chromatin remodelers as well as stable architectural changes), recent discoveries in the field of phase separation open up intriguing possibilities for how groups of enhancers are co-regulated. While there is initial evidence linking enhancers to phase-separated condensates, it is unclear how this is regulated and whether this is a general mechanism for enhancer function. Nevertheless, the concept of phase separation offers exciting possibilities for dynamic enhancer regulation and allows for quick responses to various signals. This mechanism could explain how enhancers can rapidly accumulate and induce complex gene expression programs in response to a stimulus. Simultaneously, phase separation could also create condensates that promote degradation of other proteins and transcripts to further balance the cellular transcriptome and achieve a balanced cellular composition.

Technological advances have provided a number of important avenues to utilize enhancers to understand the basic biology of the nervous system. For example, recent efforts have capitalized on the cell-type specific nature of many enhancers to drive gene expression in a highly cell-type or brain region specific manners. While previous technologies (such as BAC transgenic animals; [102]) enable population-specific gene regulation using more proximal gene promoters (up to 50 kb upstream of TSS) to drive Cre recombinase or fluorescent reporters, use of more distal enhancers as tools to label specific cell populations and/or to control expression of a transgene or endogenous genes could significantly expand these capabilities. For example, enhancer-driven gene regulation could be used to mimic complex expression patterns important for a certain tissue, developmental state or even behavior. Likewise, enhancer targeting and control over expression patterns also provides potential new approaches for cellular reprogramming. To achieve this, groups of enhancers or SEs that regulate certain expression programs in response to a stimulus or in a particular cell type could be targeted and activated.

Technological innovations have also generated useful ways to dissect enhancer function, identify mechanistic interactions and distinguish biochemical roles of specific proteins, nucleic acids and molecules (see Box 1). For example, novel CRISPR/dCas9 tools that fuse effector proteins like transcriptional activators or epigenetic modifiers to dCas9/gRNA complexes can be targeted to specific enhancer sites to induce their activity and downstream transcription profiles. Given the importance of epigenetic states to enhancer function, the use of TALEN systems or CRISPR/dCas9 constructs that anchor epigenetic modifiers at enhancers are promising tools to not only deepen our understanding of enhancer dynamics but also to target and modulate enhancer function [24,61,103,104,105,106,107]. Similarly, high-throughput reporter assays have the potential to further develop our understanding of key regulatory sequences in enhancer function, while also providing a platform for combinatorial directed evolution strategies to develop synthetic enhancers with novel properties. 

Finally, it will also be important to investigate how enhancer dynamics contribute to brain health and disease, as this provides another layer in which enhancer mediated gene regulation is vulnerable to disruption. Thus, in addition to transcriptional control as a research tool, advances in our understanding of enhancer biology could also provide promising new therapeutic avenues. Enhancer activity could potentially be targeted to correct aberrant gene expression caused by dysregulation of the enhancer (e.g., enhancer mutation) but might also be used to alleviate dysregulated gene expression in conditions with other underlying enhancer-independent mechanisms. As enhancers and particularly SEs can regulate multiple genes, targeting SEs could alleviate aberrant gene expression of entire gene programs or pathways simultaneously. Furthermore, by identifying genes that are regulated by disease-linked SEs, we could discover previously unknown elements in disease mechanisms as well novel therapeutic targets. These targets could be harnessed in the near future in several ways, including
(1)use of cell-penetrant antisense oligonucleotide strategies to modulate functional enhancer RNAs, (2)targeting enhancer-linked proteins (e.g., histone acetyltransferases) using small molecule drugs,(3)or use of enhancer sequences to drive expression in gene therapy approaches.

Additionally, future strategies could leverage human GWAS datasets with gene editing to enable pursuit of even more targeted therapeutic approaches.

Box 1Technical challenges to studying enhancers focusing on how we can identify enhancer elements, functionally validate them, screen for target genes, and functionally dissociate and study genomic enhancers and their corresponding enhancer RNAs.
**Box 1: Technical challenges to studying enhancers.**

**Identifying enhancer regions**
Based on our knowledge about the chromatin structures and epigenetic signatures of enhancer regions, ChIP-seq for these characteristic histone marks has evolved as common screening method for enhancers. Often these studies focus on H3K27 acetylation when searching for active enhancers. RNA-seq data can complement these searches and help identify active enhancers as well. One of the challenges in studying enhancers is that they typically only function in specific cell types. As a result, there are limitations when studying neuronal enhancers in many common and easy to use dividing cell lines. This also highlights the importance of single-cell techniques to study enhancers. More recently developed high-throughput ATAC-seq techniques to study chromatin accessibility in single cells have great potential to drive enhancer research forward with increased cellular resolution [24,108]. However, single cell sequencing methods that allow the assessment of the non-coding eRNAs have yet to be developed as most available technologies require polyadenylated RNA.
**Assessing enhancer function**
High throughput enhancer reporter assays such as massively parallel reporter assays (MPRAs) [109] or STARR-seq [110] have haven been valuable in large scale screens of genomic regions with enhancer function. In these assays, whole libraries of putative enhancer DNA sequences can be tested for their enhancer activity on a minimal promoter which makes it possible to test many target regions simultaneously. Even though these approaches have provided tremendous insight into the enhancer landscape, in vivo screens of enhancer activity will be the next necessary step to directly identify regions with enhancer activity in different cell types and brain regions over time and in response to different environmental stimuli. 
**Identifying enhancer targets**
While enhancers often regulate nearby genes, they can act over long distances or even regulate multiple genes [111]. It therefore remains challenging to identify target genes of enhancers. Chromatin capture sequencing approaches such as Hi-C-seq can help identify enhancer interactions with other regions in the genome and provide information about potential target genes. Unfortunately, this approach cannot currently be combined with techniques that asses functional readout (e.g., MPRAs) to be able to identify enhancer function associated with specific chromatin configurations. Technological advances that allow mapping of RNA-genome interactions (MARGI) are particularly promising in revealing eRNA interactions on a genome-wide scale. An atlas of non-coding RNA interactions could help identify previously unknown enhancer and eRNA targets and provide a deeper understanding for how non-coding RNAs affect chromatin conformation and transcriptional regulation genome-wide [112,113,114].
**Studying enhancers vs eRNAs**
Enhancer transcription is commonly used as a marker of active enhancer, which leads to the next question: How do we dissociate genomic enhancers from their enhancer RNAs when studying their function? Traditional mutations of the enhancer region do not allow us to tease apart which observed effects are based on the genomic enhancer and which are based on its eRNA. Even new genome and epigenome editing tools such as the TALEN or CRISPR-dCas9 systems, in which chromatin remodelers, and transcriptional activators or repressors can be targeted to the enhancer, cannot fully dissociate the two functional units. In addition to direct RNA knockdown approaches that don’t affect the underlying DNA, recent studies have employed CRIPSR-Display, a tool that allows for RNA tethering to the enhancer to study the effects of non-coding RNAs in a more direct fashion [60,61,103,115]. Nonetheless, studying the genomic enhancers without affecting eRNAs remains problematic.

## 4. Conclusions

Enhancers are key genomic features that not only define the development of the central nervous system but also contribute to numerous neurodevelopmental, neuropsychiatric and neurodegenerative diseases. Our understanding of enhancer function in the brain is rapidly accelerating, in part due to innovative technologies that overcome many previous limitations (Box 1). In addition to defining how enhancers work at the molecular level, this increased understanding has led to a new appreciation for the genetic substrates underlying diseases marked by enhancer dysfunction, while also providing intriguing new therapeutic targets. However, much remains unknown about enhancers. For example, technical limitations make it difficult to assign specific functional roles to all annotated enhancers—much of this work will require detailed and laborious mechanistic investigations. Likewise, the hierarchy of specific biomolecular events in enhancer activation, maintenance and decommissioning remains unclear. Finally, despite numerous breakthroughs, the extent to which enhancers can be harnessed for disease prevention or treatment is largely unknown. Thus, while previous efforts have provided the necessary first steps in dissecting enhancer activity in the genome, future work will be required to capitalize on these insights to influence human health and disease.

## Figures and Tables

**Figure 1 genes-10-00043-f001:**
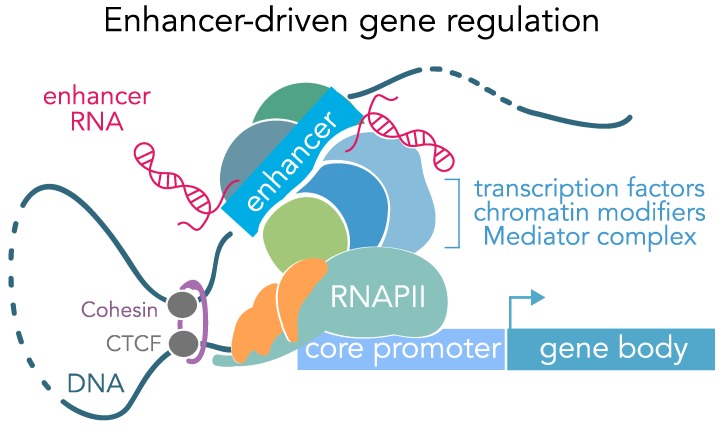
Regulation of gene expression patterns by genomic enhancers. Illustration of enhancer-promoter chromosomal looping (mediated by CTCF and cohesin) that allows distal enhancer elements to physically interact with and activate gene promoters. These interactions increase binding of transcription factors, chromatin modifiers, and the Mediator complex at gene promoters to recruit RNA polymerase II (RNAPII). Enhancers are characterized by elevated DNA sequence conservation, open chromatin, transcription factor binding motifs, characteristic histone modifications, DNA hypomethylation, and bidirectional transcription to generate enhancer RNAs (eRNAs).

**Figure 2 genes-10-00043-f002:**
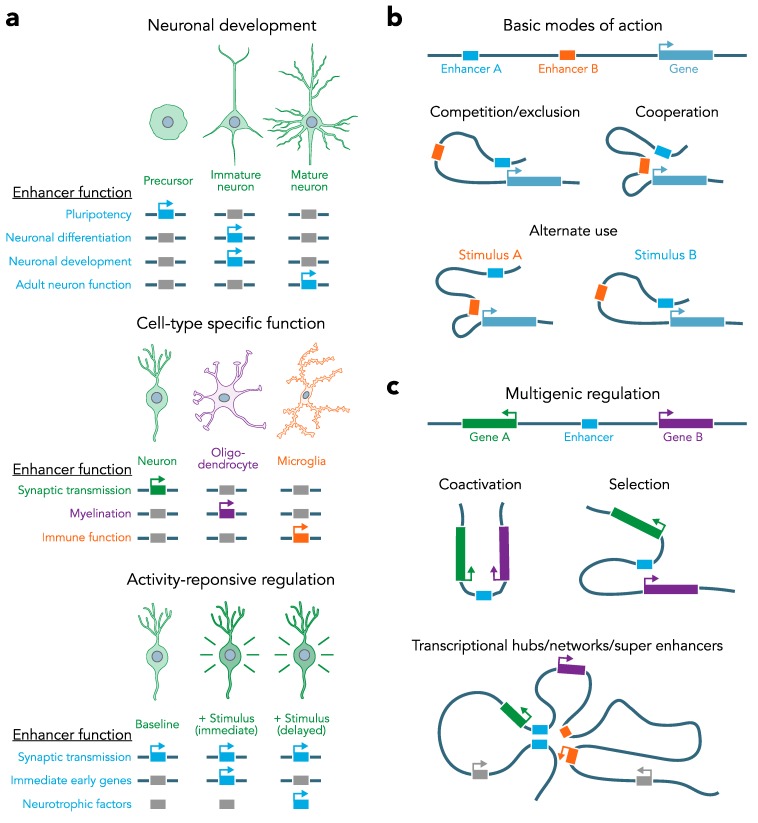
Developmental, cell-type specific, and stimulus dependent enhancer activity and looping functions. **a**, Distinct enhancers become active at specific developmental stages and in specific cell types (Top). These enhancer activity profiles regulate expression of cell identity genes in a spatiotemporal manner ensuring proper cell fate and function (Middle). A subset of enhancers responds to stimulation. These enhancers typically regulate activity-responsive genes and increase their expression in response to a certain stimulus, with specific temporal windows after stimulus induction (Bottom). **b**, Basic modes of action for enhancer looping in gene regulation. Genes can be regulated by multiple enhancers that compete with each other, act simultaneously, or alternate depending on the context or stimulus. **c**, Putative mechanisms driving coordinated regulation of many genes by single enhancers or enhancer-enhancer interactions. An enhancer can act on multiple genes in parallel, serving functions of coactivation or selection. Likewise, super enhancers or transcriptional hubs may co-localize related genes to drive expression of specific gene programs that define key cellular functions or phenotypes.

**Figure 3 genes-10-00043-f003:**
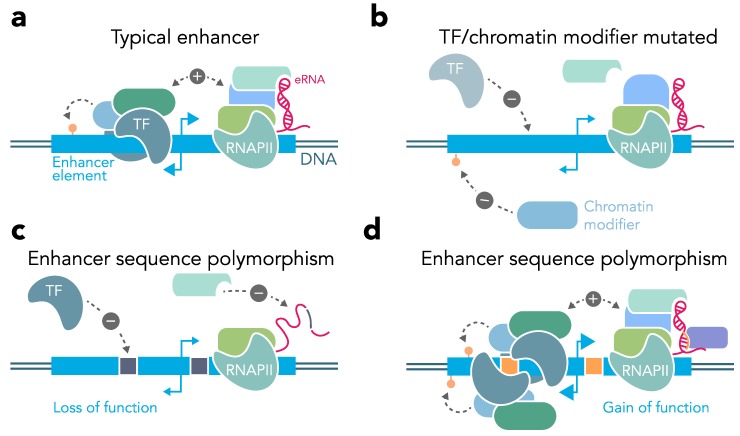
Enhancer dysfunction in genetic brain diseases and disorders. **a**, Illustration of enhancer-mediated interactions including transcription factor (TF) binding to motifs in enhancer DNA sequence, recruitment of chromatin modifying enzymes and RNA polymerase (RNAPII), as well as transcription of eRNAs. These interactions increase binding of transcription factors and epigenetic modifiers at gene promoters to regulate linked genes in cis. **b**, Mutations in chromatin modifiers and TFs commonly found at enhancers often lead to neurodevelopmental and intellectual disability. This could occur via loss of DNA binding, loss of catalytic activity, or loss of protein-ribonucleotide or protein-protein interactions that mediate enhancer function. **c-d**, Sequence polymorphisms located in enhancers could confer either loss of function (e.g., if polymorphism includes TF motif or results in change to eRNA sequence that abolishes protein-ribonucleotide interactions), or potentially gain of function (e.g., via recruitment of new TFs or novel eRNA-protein interactions).

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
