# Peer review of "Genomic Enhancers in Brain Health and Disease"

_genes, 2019, doi:10.3390/genes10010043_

Round 1

Reviewer 1 Report

The review article by Carullo and Day revises some aspects of genomic enhancers in brain health and disease. The manuscript is mainly well organized being focused on key role of enhancers as transcriptional regulators.

Major points:

1) as mentioned by the authors there several excellent reviews on the general characterization of enhancers and neuronal chromatin remodeling, the purpose of present paper being focused only on the role of the enhancers in the nervous system, as well as discussion on enhancer function on both healthy and neurological disorders associated with dysregulation of enhancers.

Consequently, I was expecting on more information or even hypotheses on mechanisms and dynamics of enhancer activities instead of listing only general aspects.

2) technological and therapeutic goals of how enhancers can be targeted are insufficiently presented in my opinion.

Author Response

Reviewer 1:

The review article by Carullo and Day revises some aspects of genomic enhancers in brain health and disease. The manuscript is mainly well organized being focused on key role of enhancers as transcriptional regulators.

Major points:

1) as mentioned by the authors there several excellent reviews on the general characterization of enhancers and neuronal chromatin remodeling, the purpose of present paper being focused only on the role of the enhancers in the nervous system, as well as discussion on enhancer function on both healthy and neurological disorders associated with dysregulation of enhancers.

Consequently, I was expecting on more information or even hypotheses on mechanisms and dynamics of enhancer activities instead of listing only general aspects.

Our response: We want to thank the reviewer for this comment. While we did spend a portion of our manuscript outlining general aspects of enhancer function, we felt this was necessary to be able to build on this information in the second half of the manuscript. Nevertheless, we have expanded the second (neurobiology-specific) section of the Review and discussed several additional studies relating enhancer function to Alzheimer’s disease, Parkinson’s disease, and Fragile X syndrome. We believe that these additions strengthen the manuscript and further highlight the potential impact of enhancer function in the brain.

Regarding the incorporation of new hypotheses into our review, we would argue that our focus on how mutations could alter enhancer dynamics to give rise to brain diseases is itself a hypothesis built on our understanding of enhancer function. We have also taken the opportunity to speculate on potential mechanisms and future areas of interest in our “Future directions” section. Together with exploration of these concepts in Figure 2 and Figure 3, we feel that this breaks ample ground for a “review” manuscript which is ostensibly geared towards summarizing the recent literature. We have highlighted as many examples of brain-specific enhancer function as is possible, but we note that at this point this is still a relatively new field, and so we do not have the benefit of decades of research into this area. 

2) technological and therapeutic goals of how enhancers can be targeted are insufficiently presented in my opinion.

Our response: We appreciate the reviewer’s comment. To address this concern, we expanded and reworked our “Future directions” section to increase coverage of these important topics. We elaborated on potential roles of enhancer in response to various stimuli, potential mechanisms by which phase separation could affect enhancer function and the transcriptome, technological opportunities including epigenetic targeting and SE targeting to manipulate enhancer function, and enhancer manipulation as a therapeutic avenue.

Author Response

Reviewer 2:

Carullo and Day present a compelling and comprehensive view on enhancer biology and its relevance to neurobiological function and disease. The review is well written, with a clear imperative to highlight the complexity of enhancer function and the emerging technologies that can be applied to its study. The figures in the manuscript are extremely well designed, and greatly enhance readability. The manuscript reflects the authors’ expertise, and provides an appropriate level of depth and timeliness of the cited studies.

The major comments below are intended to improve clarity and comprehensiveness of the manuscript, and should be addressed only if they fall within the original scope and organization intended by authors. The minor comments reflect the fact the manuscript requires copy editing and formalization of the nomenclature.

MAJOR COMMENTS

    Organization: The authors have selected to provide an overview of enhancer function and related proteins separately from the sub-section entitled, ‘Enhancer function in the brain’. This initial, 8-section, overview does directly discuss neurobiology, yet in some cases reference to neurobiology is included at the end of the section (e.g. Activity dependent enhancers (after LINE 178), SEs (LINE 192 – 194)). To improve overall organization and clarity, please (1) insert a sub-heading after the introduction that indicates general overview, not with respect to neurobiology, and (2) refer to neuro sub-section consistently throughout sections 2-9.

Our response: We want to thank the reviewer for this comment. We clarified the structure of the manuscript by dividing “General enhancer properties and functions“ and a “Brain specific enhancer function” sections, and specified in some of the sub-headings their relevance to the brain specific section.

    LINE 87: The interaction between enhancer and promoter is very interesting, and a discussion of the blurring of their distinctions is timely. To improve clarity, insert 1-2 sentences defining canonical promoters, and some aspects that classically distinguish promoters and enhancers (location relative to TSS, +/- TATA, etc). This will underscore the importance of this section. The promoter definition may also appear in the FIGURE 1 legend.

Our response: We appreciate the reviewer’s comment and we added a short definition of canonical promoters in the main text. We decided not to add this definition to the figure legend as we find that level of detail about promoter structure might confuse the reader, especially since the figure is focusing on the enhancer and not showing the promoter specific elements.

    LINE 93, Distinction between enhancers and promoters can be illustrated with a description of how Dao et al distinguished between enhancers and promoters, given the screen for ‘regions... that increase transcription,’ a feature of both.

Our response: We agree that this addition highlights the differences between promoters and enhancers and added a brief discussion of the Dao et al manuscript in this paragraph.

    LINES 221 – 241: A discussion of phase separation is timely and welcome in the neuroscience community, as many readers may not be familiar with this emerging field.

o LINES 229-241: This section defines phase separation clearly and should be moved to the beginning of the section, before LINE 221. Also, super enhancers should be abbreviated to SE, for consistency.

Our response: The SE abbreviation has been corrected, and we have reworked this section to improve clarity.

o LINES221-225:MovethissectionafterLINE241.Additional detailswould improve this section: (1) detail on the mechanism of ‘disrupting condensates,’ to distinguish this from other methods implicating coactivators in SE function (2) define ‘compartmentalization’ with respect to condensates in general and SEs in particular.

Our response: We thank the reviewer for these comments. We revised and restructured the phase separation paragraph. We define phase separation in more detail early in the paragraph and added information on what phase separated condensates are, as well as on how Sabari et disrupted such condensates using 1,6-Hexanediol.

    LINE 256: Given that one challenge in enhancer biology is defining enhancer-gene associations, please add details on the methods by which Nord et al determined that ‘dynamic H3K27 acetylation changes throughout development correspond to expression changes’ and clarify if this refers to eRNA or mRNA expression.

Our response: We appreciate the reviewer’s comment. We explained the techniques Nord et al used in this manuscript in a little more detail, highlighting their lacZ reporter assay.

    LINES 300-302, LINES 503-505: Novel approaches for enhancer-mediated gene regulation should also include work on locus-targeted epigenetic editing. See, Mendenhall

 (Bernstein) et al, Nature biotechnology 2013, Locus-specific editing of histone modifications at endogenous enhancers

Our response: We expanded on different techniques for genetic and epigenetic editing in our future directions section. We use the Mendenhall manuscript there as an example for the TALEN mediated epigenetic editing of enhancer regions to manipulate and study enhancer function.

    LINE 332: Are enhancers the only kind of ‘non-coding transcribed element’? In other words, did this study use hPTMs, TFs or other markers to specifically define enhancers in addition to their transcriptional activity?

Our response: We thank the reviewer for point out the lack of clarity. We revised this section and expanded on which characteristics were used in this manuscript and how enhancer regions were identified. We believe that this additional information clarifies which portion of non-coding transcribed elements were studied here and classified as enhancers.

    LINE 337: Additional details are needed to connect this section with earlier comments on chromatin structure of enhancers. What hPTMs does PRDM16 catalyze and are these canonical marks of enhancers? Did the study analyze the enrichment of these hPTMs or only that of PRDM16?

Our response: As a histone methyltransferase, PRDM16 has been suggested to deposit H3K9me1 and H3K4me1 mark. The discussed paper specifically measured H3K4me, a mark commonly found at active and poised enhancers.

    LINES 462-466: See, Sun (Cremins) et al, Cell 2018, Disease-Associated Short Tandem Repeats Co-localize with Chromatin Domain Boundaries.

Our response: We thank the reviewer for suggesting to include this citation in our review. We added a paragraph to the “mutations of enhancer loci” section discussing the importance and implications of this work for how variants can disrupt TAD boundaries, impair CTCF binding, and ultimately disrupt enhancer promoter looping and aberrant transcription

    LINES 496-505 (Also see LINES 427-428): The discussion of enhancers as methods to control cell type specific gene expression would be more complete with direct reference to BAC transgenic technology. In this case, up to 50kB of sequence upstream from a given TSS is used to drive reporter or Cre expression (See Gong (Heintz) et al, Nature 2003, A gene expression atlas of the central nervous system based on bacterial artificial chromosomes). While this technology is not new, failure to mention it implies that enhancer-based gene regulatory methods are not yet established.

Our response: We revised this paragraph to refer to previously developed BAC transgenic models to avoid confusion about the novelty and availability of enhancer-based gene editing models.

    LINES 506-510: Authors may choose to site recent work identifying therapeutic potential of enhancers (See, for example, Sanchez-Mut (Graff) et al, Nature medicine 2018, PM20D1 is a quantitative trait locus associated with Alzheimer’s disease

Our response: We appreciate the reviewer’s comment. We included a discussion of this relevant work in the “mutations of enhancer loci” section as we believe if allows us to expand on this in more detail in that section.

    FIGURES: The figures in this manuscript are excellent and contribute greatly to the manuscript. In particular, Figure 2 is a brilliant summary of enhancer function and the important roles of enhancers in neurons. Some minor changes will improve clarity.

o FIGURE2c: Is the bottom schematic illustrating a super enhancer? If so, can this be labeled?

o FIGURE3b: Is panel missing a label for the chromatin modifier? The mutated TF binding site is clear, but it is not clear what has changed for the other protein that is illustrated as ‘off’ the complex.

Our response: Thank you for pointing out that additional labels are needed for more clarity. We added labels to both figure to avoid confusion.

MINOR COMMENTS

Our response: We appreciate the reviewer’s comments and thank them for catching these minor issues. We addressed all minor comments in our manuscript.

    LINE 68: Insert reference on estimates of number of enhancers

    LINE 82: Term ‘bi-directional transcription’ is confusing, since enhancers themselves are non-coding. Please revise for clarity.

    LINE 129: Define ATAC-seq acronym

    Correct double space to single space throughout (e.g. LINES 91, 130, 156, 297, 396, 397, 431, 435, 474 etc)

    Please consult with editors on use of correct gene/protein nomenclature. (1) Protein names (e.g Mediator, Nanog) are capitalized throughout (2) most proteins are presented in acronym only, without full protein name provided (3) protein names are written in all capital letters, while references may cite non-human studies.

    Figure 2 appears at LINE 197, but reference to Figure 2 doesn’t appear until LINE 354

    Figure 3 appears at LINE 306, but reference to Figure 2 doesn’t appear until LINE 413

    LINES 204-206: Include citation that defines eRNAs.

    LINE 221: Missing closing parenthesis after [60]

    LINE 252: Define acronym ChIP-seq

    LINE 254: Missing the word ‘by’ after ‘driven’

    LINE 261: Cite reference for ‘lead SNP for depression and alcohol dependence’

    LINE 299: correct spelling – approaches

    LINE 317: delete ‘distinct’

    LINES 365-375: revert to past tense for consistency with rest of manuscript

    LINE 390: delete second instance of ‘release’

    LINE 394: missing ‘for’ after ‘important’

    LINE 416: abbreviate transcription factors to TFs for consistency

    BOX 1: correct ‘Identifying enhancers region’ to ‘Identifying enhancer region’
